# Ecological versatility and biotechnological promise: Comprehensive characterization of the isolated thermophilic *Bacillus* strains

**Hazem Aqel**[1]*, **Husni Farah**[2], **Afnan Al-Hunaiti**[3]

**1** Basic Medical Sciences Department, Al-Balqa' Applied University, Salt, Jordan, **2** Medical Laboratory Sciences Department, Al-Ahliyya Amman University, Amman, Jordan, **3** Chemistry Department, Jordan University, Amman, Jordan

☉ These authors contributed equally to this work.
* hazem.aqel@bau.edu.jo

**Data Availability Statement:** Data relevant to this study are available from Dryad at doi:10.5061/dryad.dncjsxm5v.

## Abstract

This study focuses on isolated thermophilic *Bacillus* species' adaptability and physiological diversity, highlighting their ecological roles and potential industrial applications. We specifically investigated their capacity to thrive in extreme conditions by examining their environmental tolerances and adaptations at the metabolic and genetic levels. The primary objective is to evaluate the suitability of these species for biotechnological applications, considering their resilience in harsh environments. We conducted a comparative analysis of the environmental adaptability parameters for various *Bacillus* species. This included examining growth temperature ranges, pH tolerance, oxygen requirements, carbohydrate fermentation patterns, colony morphology, enzymatic activities, and genetic properties. Controlled laboratory experiments provided the data, which were then analyzed to determine patterns of adaptability and diversity. The research revealed that *Bacillus* species could endure temperatures as high as 73°C, with a generally lower growth limit at 43°C. However, strains TBS35 and TBS40 were exceptions, growing at 37°C. Most strains preferred slightly alkaline conditions (optimal pH 8), but TBS34, TBS35, and TBS40 exhibited adaptations to highly alkaline environments (pH 11). Oxygen requirement tests classified the species into aerobic, anaerobic, and facultative aerobic categories. Genetic analysis highlighted variations in DNA concentrations, 16s rRNA gene lengths, and G+C content across species. Although glucose was the primary substrate for carbohydrate fermentation, exceptions indicated metabolic flexibility. The enzymatic profiles varied, with a universal absence of urease and differences in catalase and oxidase production. Our findings underscore thermophilic Bacillus species' significant adaptability and diversity under various environmental conditions. Their resilience to extreme temperatures, pH levels, varied oxygen conditions, and diverse metabolic and genetic features emphasize their potential for biotechnological applications. These insights deepen our understanding of these species' ecological roles and highlight their potential industrial and environmental applications.

**Funding:** The author(s) received no specific funding for this work.

**Competing interests:** The authors have declared that no competing interests exist.

## Introduction

Bacteria of the genus *Bacillus* are found widely in different environments and exhibit diverse metabolic capabilities and the ability to adapt to extreme conditions. The thermophilic *Bacillus* strains are particularly interesting because they can survive in high-temperature environments and have potential uses in various biotechnological applications [1,2].

Previous studies have investigated the physiological and metabolic properties of thermophilic *Bacillus* species. They found that these bacteria are adaptable regarding using different substances, producing enzymes, and withstand stressful conditions. This research has provided a basis for understanding the ecological functions of these bacteria and their potential involvement in industrial processes, bioremediation, and the production of bioactive compounds [3–5]. Although significant progress has been made, a more comprehensive understanding of the isolated thermophilic *Bacillus* strains is possible [6].

The main objective of this study was to extend previous research and analyze thermophilic *Bacillus* strains that have been isolated on a large scale. We focused on understanding their growth conditions, oxygen requirements, colony appearance, and enzyme profiles [7,8].

By carefully studying the physical properties and metabolic capabilities of thermophilic *Bacillus* species, we can add to the growing body of information on this topic. The results of this research could improve our understanding of microbial diversity and initiate new scientific and industrial [1,4].

Although understanding the properties and uses of thermophilic *Bacillus* strains has improved significantly, there are still some areas where our knowledge needs to be improved and could be improved. Previous research has shed light on their metabolic capabilities and resistance to stress. However, we still need to fully explore their ecological adaptability and uncover their potential applications in biotechnology [9,10]. Further studies are required to explore the diversity of enzymatic activities of these isolated strains and how they relate to their ability to adapt to different environments [5].

The main aim of this study was to thoroughly investigate and describe isolated thermophilic *Bacillus* strains, focusing on their ability to undergo diverse metabolism, adapt to different environments, and produce enzymes. Using a multifaceted approach, we aimed to uncover the complexity of their physiological and metabolic properties and provide valuable information about their ecological roles and potential uses in biotechnology [5,11]. This comprehensive study is expected to improve and expand our knowledge of microbial diversity within thermophilic *Bacillus* species, leading to advances in microbial biotechnology and environmental protection [9].

## Materials and methods

### Isolation of thermophilic Bacillus strains

Thermal waters were collected from the Ma'in Hot Spring, Jordan. Water samples were collected in 500 mL empty sterile conical flasks and 500 mL sterile conical flasks containing 100 mL of peptone yeast extract broth. The collected samples were transported directly to the laboratory for immediate testing [12].

Water samples were streaked on Thermus agar (ATCC medium 697) and incubated at 43°C for 48 h and anaerobically. Different colonies were picked and subcultured on the same medium and under different environmental conditions to obtain a pure culture for each colony. All picked colonies grew at 43°C and were incubated at the highest temperatures to examine their tolerance at different temperatures. An extremely well-known *Bacillus* strain was used as a control: *Bacillus stearothermophilus* (ATCC12980).

In conducting our research at the Ma'in Hot Springs, we did not require any specific permits. This is due to the fact that the Ma'in Hot Springs are a publicly accessible location, open to all without the need for special permission or access grants. The nature of our study did not involve any activities that would typically necessitate permits, such as environmental disturbance or the collection of protected species. Our research activities were confined to observation, data and water collection that complied with the general public use guidelines of the site. Therefore, our study was conducted in full compliance with the regulations governing the use of the Ma'in Hot Springs, without the need for additional permits or approvals from authorities.

## Morphological identification

Morphological properties were investigated from 18 to 24 h in bacterial cultures grown in tryptone dextrose agar (TDA) plates. These included Gram stain reaction and spore-position [9,13]. The morphology of bacterial colonies on solid media and bacterial growth in the broth were also observed.

## Assessment of the growth parameters

The ability of isolated thermophilic *Bacillus* strains to grow at different temperatures (10, 25, 37, 43, 53, 63, 73, and 83oC) and different pH values (2, 3, 4, 5, 6, 7, 8, 9, 10, and 11) was assessed in peptone yeast extract agar. Each isolate's minimal, optimal, and maximal temperature and pH values were determined [10,13,14].

## Determination of the oxygen dependency

Peptone yeast extract agar was cultivated under distinct oxygen conditions, including aerobic, anaerobic, and microaerophilic environments. Comparative growth rates were assessed through optical density measurements, facilitating the determination of oxygen preferences for each isolate [1,15].

## Evaluation of enzyme production and substrate hydrolysis

The isolates were tested for various enzymatic activities, including catalase, oxidase, and urease. Substrate and hydrolysis tests were performed using media supplemented with specific substrates such as casein, starch, gelatin, and citrate [4,11,13].

Starch hydrolysis was tested on peptone yeast extract agar, replacing sucrose with 10 g of soluble starch per liter. A clear zone around the colony indicated a positive reaction after flooding with Gram's iodine solution. Gelatin hydrolysis was tested on tubes containing a solid gelatin medium. A positive result was fluid production due to the bacterial enzyme gelatinase. Casein hydrolysis was tested on peptone yeast extract agar containing 10 g glucose per liter and 25 mL 15% (w/v) skim milk per liter and adjusted to pH 5.6. A positive result was a clear zone around a strip inoculation after flooding with 10% (w/v) $HgCl_2$.

## Carbohydrate fermentation profiling

Carbohydrate fermentation tests were conducted using a variety of carbohydrates, including glucose, lactose, mannose, inositol, sorbitol, rhamnose, saccharose, melibiose, amylose, and arabinose. The production of acid and gas was monitored to determine the use of each carbohydrate [16].

## Conduct additional metabolic and tolerance tests

Additional tests were performed to evaluate the isolates' metabolic capabilities and environmental adaptability. These included tests for salt tolerance (7% NaCl), indole production (IND), Voges-Proskauer test (VP), nitrate reduction ($NO_2$), O-nitrophenyl-β-D-galactopyranoside test (ONPG), nitrogen gas production (N2 gas), and tests for arginine dihydrolase (ADH), ornithine decarboxylase (ODC), and lysine decarboxylase (LDC) [8]

## DNA extraction

Cellular DNA was obtained from each *Bacillus* colony grown overnight in LB broth culture using a DNA Extraction Kit (QIAamp DNA Mini blood kits, Quigen) following the manufacturer's instructions. Ten microliters of overnight LB broth culture in a tube were centrifuged at 8000 rpm for 10 min, and the supernatant was discarded. The pellet was re-suspended in phosphate buffer saline using a vortex, centrifuged again, and the supernatant was discarded. The pellet was resuspended in 2 mL of TE buffer and mixed by vortex. 500 μL was transferred to 1.5 mL Eppendorf, and 50 μL of SDS (10%) and 5 μL of proteinase K were added and left in the incubator for 1.5 h at 37°C. An equal volume of (555 μL) phenol, chloroform, and isoamyl mixture was added, vortexed for 1 min, centrifuged for 4 min, and the aqueous supernatant containing DNA was transferred to another tube. Fifty-five μL of 3 M sodium acetate and a double volume of cold alcohol (100%) were added, mixed gently 2–3 times, and centrifuged for 10 min. The supernatant was removed, and the pellet was washed gently with 1 mL of 70% alcohol and centrifuged for 1 min. The supernatant was removed, the pellet was left to dry, resuspended in 10 μL of TE buffer, incubated for 30 min at 55°C, and stored at -20°C. The extracted DNA for each tested bacterium was calculated using a NanoDrop 8000 spectrophotometer (Thermoscientific).

## Genetic profiling

The guanine-cytosine (GC) content was determined using the thermal denaturation method. This provided insight into the genetic composition of the isolates [17].

## Detection of 16s rRNA using PCR

All tested *Bacillus* species were tested for the presence of 1500–1700 bp PCR product of 16s rRNA using the following primers: forward (5'-AGAGTTTGATGATCCTGGCTCAG-3') and reverse (5'-GGTTACCTTGTTACGACTT-3'). The amplification reaction was carried out in 25 μL volumes, following the conditions in Table 1. The results of 16s rRNA were observed by using 2% agarose gel electrophoresis.

All experiments were conducted in triplicate, and the results were averaged to ensure accuracy. The equipment and materials used were regularly sterilized to prevent contamination.

**Table 1. Primers were used in 16s rRNA PCR.**

| Primer name | Oligonucleotide sequence (5'-3') | Primer size (bp) | PCR condition | Reference |
|---|---|---|---|---|
| 8–27 F<br>1492 R | 5'-AGAGTTTGATGATCCTGGCTCAG-3'<br>5'-GGTTACCTTGTTACGACTT-3' | 1500–1600 | 96°C → 2 min<br>96°C → 10 sec<br>55°C → 5 sec<br>72°C → 45 sec }25X<br>60°C → 4 min<br>4°C→∞ | [18] |

## Results

### Comparative analysis of the growth temperature ranges of isolated thermophilic *Bacillus* species

Table 2 shows the variations in ideal growth temperatures between the different thermophilic *Bacillus* species. However, it is also emphasized that all these species tolerate a maximum growth temperature of 73˚C. Most of these species have a similar lowest temperature at which they can grow, which is 43˚C. However, two species, TBS35 and TBS40, tolerate a slightly lower minimum temperature of 37˚C. Despite this difference in minimum growth temperature, all species mentioned can thrive up to a maximum temperature of 73˚C. This growth indicates a universal upper limit to their ability to tolerate high temperatures.

However, these species vary in their ideal growing temperature. B. stearothermophilus thrives best at 56˚C, while TBS5 and TBS35 prefer slightly higher temperatures of 58˚C and 60˚C, respectively. Similarly, TBS40 thrives at an optimal growth temperature of 60˚C.

### Diversity in growth pH tolerance among isolated thermophilic *Bacillus* species

Values indicate the pH range in which these species survive and thrive. These pH values include the lowest, most favorable, and highest. Most species, such as *B. stearothermophilus*, TBS4, TBS25, TBS26, TBS55, and TBS109, have a minimum growth pH of 5 and an optimal growth pH of 8.

However, there are significant differences between the members of this group. TBS5 is characterized by growing best at a pH of 6, which is more acidic than the pH of 8, which most species prefer. On the other hand, TBS6 shows a different characteristic with a higher minimum growth pH of 7, indicating its inability to flourish in acidic environments, similar to its fellow members.

A special subset in the table includes TBS34, TBS35, and TBS40. These three species exhibit the same growth pH pattern, with their minimum, optimal, and maximum pH values of 11. This pH indicates that these species are specifically adapted to thrive in extremely alkaline conditions and cannot withstand acid.

The third table presents the variation in pH tolerance and preference among the thermophilic Bacillus species that have been isolated. Although many have similar growth pH values,

**Table 2. Variability in minimum, optimal, and maximum growth temperatures among thermophilic *Bacillus* isolates.**

| Thermophilic *Bacillus* species | Temperature (˚C) | | |
|---|---|---|---|
| | Minimum | Optimal | Maximum |
| *B. stearothermophilus* | 43 | 56 | 73 |
| **TBS4** | 43 | 50 | 73 |
| **TBS5** | 43 | 58 | 73 |
| **TBS6** | 43 | 51 | 73 |
| **TBS25** | 43 | 54 | 73 |
| **TBS26** | 43 | 51 | 73 |
| **TBS34** | 43 | 54 | 73 |
| **TBS35** | 37 | 60 | 73 |
| **TBS40** | 37 | 60 | 73 |
| **TBS45** | 43 | 55 | 73 |
| **TBS55** | 43 | 52 | 73 |
| **TBS109** | 43 | 53 | 73 |

suggesting a general preference for alkaline conditions, certain species exhibit distinctive adaptations, highlighting the ecological diversity within this group of bacteria.

## Diversity of oxygen requirement for growth of isolated thermophilic *Bacillus* species

The oxygen requirements of these species are divided into three specific types in the table: GSPO (Thrives with oxygen—Aerobic), GSAO (Thrives without oxygen—Anaerobic), and GSPAO (Thrives both with and without oxygen—Facultative aerobic). Most species, particularly *B. stearothermophilus*, TBS4, TBS6, TBS34, TBS55, and TBS109, fall into the GSAO category. This classification indicates that they thrive in conditions without oxygen.

Certain species, such as TBS5, TBS25, TBS26, TBS35, and TBS40, belong to different groups known as GSPAO. These species prefer facultative aerobic conditions. This facultative aerobic means they can adapt to their oxygen requirements and thrive in environments with or without oxygen. Their ability to adapt their metabolism to oxygen availability allows them to thrive in various habitats.

TBS45 stands out among the species listed in the table because it belongs to the GSPO category, indicating its robust growth under aerobic conditions. This aerobic condition emphasizes TBS45's unique ability to adapt and thrive in environments abundant with oxygen, enabling it to efficiently generate energy through aerobic respiration.

Table 3 shows the varying oxygen requirements of the isolated thermophilic *Bacillus* species. These needs range from the ability to survive without oxygen to the ability to adapt to both oxygenic and oxygen-free environments and the need for oxygen to survive.

## Differences in colony morphology, broth growth, and hemolytic properties of isolated thermophilic *Bacillus* species

Table 4 details the colony growth characteristics of several isolated *Bacillus* species, including *B. stearothermophilus*, TBS4, TBS5, TBS6, TBS25, TBS26, TBS34, TBS35, TBS40, TBS45, TBS55, and TBS109. The characteristics under observation included colony morphology, broth growth, and hemolysis type. Nevertheless, *B. stearothermophilus* exhibits a difference as its colonies appear in a white-cream hue. Notably, TBS35 and TBS40 display colorless colonies, whereas TBS45 showcases colonies in yellow. Regarding the shape of the colonies, most species exhibit circular formations.

**Table 3. Comparative study of minimum, optimal, and maximum growth pH-values for thermophilic *Bacillus* isolates.**

| Thermophilic *Bacillus* species | pH-value | | |
|---|---|---|---|
| | Minimum | Optimum | Maximum |
| *B. stearothermophilus* | 5 | 8 | 11 |
| **TBS4** | 5 | 8 | 11 |
| **TBS5** | 5 | 6 | 11 |
| **TBS6** | 7 | 8 | 11 |
| **TBS25** | 5 | 8 | 11 |
| **TBS26** | 5 | 8 | 11 |
| **TBS34** | 11 | 11 | 11 |
| **TBS35** | 11 | 11 | 11 |
| **TBS40** | 11 | 11 | 11 |
| **TBS45** | 6 | 8 | 11 |
| **TBS55** | 5 | 8 | 11 |
| **TBS109** | 5 | 8 | 11 |

**Table 4. Comparative analysis of aerobic and anaerobic growth preferences in thermophilic *Bacillus* isolates.**

| Thermophilic *Bacillus* species | Oxygen requirement | | |
|---|---|---|---|
| | GSPO | GSAO | GSPAO |
| *B. stearothermophilus* | No | Yes | No |
| TBS4 | No | Yes | No |
| TBS5 | Yes | Yes | Yes |
| TBS6 | No | Yes | No |
| TBS25 | Yes | Yes | Yes |
| TBS26 | Yes | Yes | Yes |
| TBS34 | No | Yes | No |
| TBS35 | Yes | Yes | Yes |
| TBS40 | Yes | Yes | Yes |
| TBS45 | Yes | No | No |
| TBS55 | No | Yes | No |
| TBS109 | No | Yes | No |

GSPO: Grow strongly in the presence of oxygen (aerobic); GSAO: Grow strongly in the absence of oxygen (Anaerobic); GSPAO: Grow strongly in the presence and absence of oxygen (Facultative aerobic).

Different elevations can be observed among the various species. *B. stearothermophilus*, TBS4, TBS55, and TBS109 display raised colony growth with a convex shape. On the other hand, TBS34, TBS35, and TBS40 stand out by forming colonies with a flat elevation. When looking at the edges of the colonies, most species have smooth boundaries without any breaks.

When observing the way different species grow in broth culture, it is evident that certain species, such as *B. stearothermophilus*, TBS4, TBS6, TBS34, TBS55, and TBS109, display sediment at the bottom of the culture called SAB. Conversely, TBS5, TBS25, TBS26, TBS35, and TBS40 exhibit dispersed growth throughout the broth, leading to a cloudy appearance.

The information provided by the table also reveals the varied hemolytic characteristics exhibited by these *Bacillus* species. *B. stearothermophilus*, TBS4, TBS26, TBS35, TBS40, and TBS45 were characterized by putative gamma-hemolysis, referred to as γ. In contrast, TBS5, TBS6, TBS25, TBS34, TBS55, and TBS109 showed a different type of hemolysis, presumably beta-hemolysis, referred to as β.

Table 4 comprehensively shows various thermophilic Bacillus species' colony growth characteristics and hemolytic properties. This hemolysis highlights the wide range of morphological and physiological traits exhibited by this particular group of bacteria.

## Morphological and cellular diversity among the isolated thermophilic *Bacillus* species as revealed by Gram staining and wet mount analysis

Table 5 comprehensively analyzes the traits exhibited by various thermophilic *Bacillus* species, as observed using Gram-staining and wet mount methods. Bacillus species display a variety of shapes. TBS45 stands out with its short and slender rods, indicating a difference in cell width compared with the rest of the species.

The organization of the cells also differs among different species. TBS45 exhibits a more intricate organization in which cells form distinct, paired, and linked chains, suggesting its desire to create elongated formations.

Another observed feature is the spore position within the cells. *B. stearothermophilus* exhibits a variety of spore positions, with central, subterminal, and terminal positions indicated. TBS4, TBS6, TBS25, TBS26, and TBS45 have terminal spore positions, while TBS5, TBS34,

**Table 5. Comparative characterization of colony growth and hemolytic traits in thermophilic *Bacillus* isolates.**

| Thermophilic *Bacillus* species | Colony morphology | | | | Broth growth | Type of hemolysis |
|---|---|---|---|---|---|---|
| | Color | Form | Elevation | Margin | | |
| **B. stearothermophilus** | White | Circular | Convex | Entire | SAB | Y |
| **TBS4** | White | Circular | Convex | Entire | SAB | Y |
| **TBS5** | White | Circular | Raised | Entire | Turbidity | β |
| **TBS6** | White | Circular | Raised | Entire | SAB | β |
| **TBS25** | White | Circular | Raised | Entire | Turbidity | β |
| **TBS26** | White | Circular | Raised | Entire | Turbidity | Y |
| **TBS34** | White | Irregular | Flat | Lobate | SAB | β |
| **TBS35** | Colorless | Irregular | Flat | Curled | Turbidity | Y |
| **TBS40** | Colorless | Irregular | Flat | Curled | Turbidity | Y |
| **TBS45** | Yellow | Circular | Raised | Entire | Layer at the top | Y |
| **TBS55** | White | Circular | Convex | Entire | SAB | β |
| **TBS109** | White | Circular | Convex | Entire | SAB | β |

SAB: Sediment at bottom; Y -hemolysis: Gamma hemolysis (non-hemolytic- no zone); β-hemolysis: Beta hemolysis (destruction red blood cells- clear zone).

TBS35, and TBS40 have subterminal spores. In TBS55 and TBS109, the spores are located in the center of the cells (S1–S10 Figs and S1 Table).

The spore morphology of the species is generally oval, as observed in *B. stearothermophilus*, TBS4, TBS5, TBS34, TBS35, TBS40, TBS45, TBS55, and TBS109. However, TBS6, TBS25, and TBS26 exhibit round spore morphology (S1–S10 Figs).

Swelling of the bacillus body is another feature observed. Most species, including *B. stearothermophilus*, TBS4, TBS25, TBS26, TBS40, TBS45, TBS55, and TBS109, exhibit positive swelling of the bacillary body. In contrast, TBS5, TBS6, TBS34, and TBS35 did not exhibit swelling, indicating cellular response to sporulation differences.

Finally, motility is a universal feature of all *Bacillus* species listed. All species, including *B. stearothermophilus*, TBS4, TBS5, TBS6, TBS25, TBS26, TBS34, TBS35, TBS40, TBS45, TBS55, and TBS109, exhibit positive motility, highlighting their ability to move in their environment actively.

Table 5 details the various morphological and cellular characteristics of thermophilic *Bacillus* species, highlighting variations in cell morphology, cell arrangement, spore position and morphology, cellular swelling, and motility, which underscore the adaptability and diversity within this group of bacteria.

## Diversity of carbohydrate fermentation in isolated thermophilic *Bacillus* species

Table 6 shows the ability of several isolated thermophilic *Bacillus* species, including *B. stearothermophilus*, TBS4, TBS5, TBS6, TBS25, TBS26, TBS34, TBS35, TBS40, TBS45, TBS55, and TBS109, to ferment various carbohydrates. The carbohydrates tested were glucose (Gluc), lactose (Lac), mannose (Man), inositol (Ino), sorbitol (Sor), rhamnose (Rha), sucrose (Sac), melibiose (Mel), amylose (Amy), and arabinose (Ara).

A striking pattern in the table is the predominance of glucose fermentation among *Bacillus* species. *B. stearothermophilus*, TBS5, TBS6, TBS25, TBS26, TBS34, TBS55, and TBS109 all show the ability to ferment glucose, as the positive results indicate. This result suggests that glucose is a commonly used carbohydrate source for these thermophilic *Bacillus* species, based on the preference of many bacteria for glucose because of its primary role in cellular metabolism.

**Table 6. Comparative assessment of cell morphology, spore characteristics, and motility in isolated thermophilic *Bacillus* species.**

| Thermophilic *Bacillus* species | Gram-stain morphology | | | | | Wet mount |
| | Morphology | Arrangement of cells | Spore location | Spore morphology | Swelling of the bacillary body | Motility |
|---|---|---|---|---|---|---|
| *B. stearothermophilus* | Short Thick rods | Single & double | Central, Subterminal, &Terminal | Oval | Positive | Positive |
| **TBS4** | Long thick rods | Single | Subterminal | Oval | Positive | Positive |
| **TBS5** | Short thick rods | Single & double | Central and Subterminal | Oval | Negative | Positive |
| **TBS6** | Long thick rods | Single & double | Subterminal | Round | Negative | Positive |
| **TBS25** | Long very thick | Single & double | Subterminal | Round | Positive | Positive |
| **TBS26** | Short thick rods | Single & double | Central and Subterminal | Round | Positive | Positive |
| **TBS34** | Long thick rods | Single | Subterminal | Oval | Negative | Positive |
| **TBS35** | Short thick rods | Single, double & clusters | Subterminal | Oval | Negative | Positive |
| **TBS40** | Short Thick rods | Single & double | Subterminal | Oval | Positive | Positive |
| **TBS45** | Short thin rods | Single, double & chains | Terminal | Oval | Positive | Positive |
| **TBS55** | Short thick rods | Double | Central | Oval | Positive | Positive |
| **TBS109** | Short thick rods | Single & double | Central | Oval | Positive | Positive |

However, TBS4 and TBS45 were exceptions because they cannot ferment any listed carbohydrates, including glucose. This result indicates a potential difference in carbohydrate metabolism and use for these species compared with the other species.

TBS35 and TBS40 demonstrate different carbohydrate preferences, with both species showing the ability to ferment mannose and inositol. Additionally, TBS35 can ferment sorbitol, indicating a broader spectrum of carbohydrate utilization for this species. The positive fermentation of mannose, inositol, and sorbitol by TBS35 and TBS40 highlights their metabolic versatility and adaptation to alternative carbohydrate sources.

Most *Bacillus* species tested showed no ability to ferment lactose, rhamnose, saccharose, melibiose, amylose, or arabinose, as indicated by the negative results across these columns for most species. This result suggests that these carbohydrates are unsuitable substrates for fermentation by thermophilic *Bacillus* species, indicating specific metabolic adaptations and substrate preferences.

Table 6 provides insights into the carbohydrate fermentation capabilities of the isolated thermophilic *Bacillus* species. While glucose emerges as a commonly fermented carbohydrate among many species, variations are observed, with some species exhibiting the ability to ferment alternative carbohydrates and others showing no carbohydrate fermentation. These findings underline thermophilic *Bacillus* species' metabolic diversity and adaptability in response to different carbohydrate substrates available in their environment.

## Diversity in enzyme production and substrate hydrolysis among isolated thermophilic *Bacillus* species

The table assesses the manufacturing of three enzymes, namely catalase, oxidase, and urease, as well as the capacity of these organisms to break down casein starch. One important finding from the table was that most *Bacillus* species produce catalase and oxidase enzymes, indicating their ability to handle oxidative stress and use various electron donors. However, TBS5 and TBS45 do not produce catalase, and TBS34 does not produce oxidase. This suggests that these

**Table 7. Comparative analysis of substrate preferences in carbohydrate fermentation by thermophilic *Bacillus* isolates.**

| Thermophilic *Bacillus* species | Carbohydrates fermentation | | | | | | | | | |
|---|---|---|---|---|---|---|---|---|---|---|
| | **Gluc** | **Lac** | **Man** | **Ino** | **Sor** | **Rha** | **Sac** | **Mel** | **Amy** | **Ara** |
| *B. stearothermophilus* | Pos | Neg | Neg | Neg | Neg | Neg | Neg | Neg | Neg | Neg |
| **TBS4** | Neg | Neg | Neg | Neg | Neg | Neg | Neg | Neg | Neg | Neg |
| **TBS5** | Pos | Neg | Neg | Neg | Neg | Neg | Neg | Neg | Neg | Neg |
| **TBS6** | Pos | Neg | Neg | Neg | Neg | Neg | Neg | Neg | Neg | Neg |
| **TBS25** | Pos | Neg | Neg | Neg | Neg | Neg | Neg | Neg | Neg | Neg |
| **TBS26** | Pos | Neg | Neg | Neg | Neg | Neg | Neg | Neg | Neg | Neg |
| **TBS34** | Pos | Neg | Neg | Neg | Neg | Neg | Neg | Neg | Neg | Neg |
| **TBS35** | Neg | Neg | Pos | Pos | Pos | Neg | Neg | Neg | Neg | Neg |
| **TBS40** | Neg | Neg | Pos | Pos | Neg | Neg | Neg | Neg | Neg | Neg |
| **TBS45** | Neg | Neg | Neg | Neg | Neg | Neg | Neg | Neg | Neg | Neg |
| **TBS55** | Pos | Neg | Neg | Neg | Neg | Neg | Neg | Neg | Neg | Neg |
| **TBS109** | Pos | Neg | Neg | Neg | Neg | Neg | Neg | Neg | Neg | Neg |

Pos: Positive; Neg: Negative; glucose (Gluc); lactose (Lac); mannose (Man); inositol (Ino); sorbitol (Sor); rhamnose (Rha);sucrose (Sac); melibiose (Mel); amylose (Amy); arabinose (Ara).

*Bacillus* species may have different coping mechanisms for environmental conditions and metabolic demands.

However, none of the *Bacillus* species mentioned produced the urease enzyme. The fact that all these species cannot produce urease indicates a shared metabolic trait. This result indicates that these species do not rely on urea as a nitrogen source, which could be an important characteristic in their specific environments.

*B. stearothermophilus* and TBS109 are known for their capacity to break down both starch and gelatin, making them unique in substrate hydrolysis. This ability suggests that these species have a varied metabolic rate, allowing them to utilize various nutrients.

Table 7 shows the diverse abilities of thermophilic *Bacillus* species in producing enzymes and hydrolyzing substrates. Although catalase and oxidase production are commonly found in these species, the absence of urease production and selective substrate hydrolysis emphasize their unique metabolic diversity and specialization in adapting to their specific surroundings.

### Diversity in the metabolic capabilities and environmental adaptations of isolated thermophilic *Bacillus* species

The test outcomes offer valuable information about metabolic abilities, the ability to withstand salty conditions, and possible enzyme functions. Upon examining Table 8, it becomes clear that the *Bacillus* species display diverse tolerance toward 7% NaCl.

While most species demonstrate an inability to thrive in highly saline conditions, there were three exceptions: TBS35, TBS40, and TBS109. These species yielded positive results, indicating a higher capacity to withstand salt. This unique characteristic indicates they can occupy more diverse and demanding environments than their counterparts.

These findings indicated differences in indole production (IND) among species. TBS4, TBS5, TBS26, and TBS55 are recognized as indole producers, indicating their ability to break down tryptophan. This ability hints at their potential to use various amino acids for metabolic activities. The Voges-Proskauer test (VP) results demonstrated that TBS5, TBS6, TBS55, and TBS109 can generate acetoin, a metabolic compound. The capacity to decrease nitrate implies that these organisms can engage in nitrogen cycling in their surroundings. The production of

**Table 8. Comparative analysis of metabolic capabilities: Enzyme production and hydrolysis substrates in thermophilic *Bacillus* isolates.**

| Thermophilic *Bacillus* species | Enzyme production | | | Substrate hydrolysis | | | |
|---|---|---|---|---|---|---|---|
| | Catalase | Oxidase | Urease | Casein | Starch | Gelatin | Citrate |
| *B. stearothermophilus* | Pos | Pos | Neg | Neg | Pos | Pos | Neg |
| **TBS4** | Pos | Pos | Neg | Neg | Neg | Neg | Neg |
| **TBS5** | Neg | Pos | Neg | Neg | Neg | Neg | Neg |
| **TBS6** | Pos | Pos | Neg | Neg | Neg | Neg | Neg |
| **TBS25** | Pos | Pos | Neg | Neg | Neg | Neg | Neg |
| **TBS26** | Pos | Pos | Neg | Neg | Neg | Neg | Neg |
| **TBS34** | Pos | Neg | Neg | Neg | Neg | Neg | Neg |
| **TBS35** | Pos | Pos | Neg | Neg | Neg | Neg | Neg |
| **TBS40** | Pos | Pos | Neg | Neg | Neg | Neg | Neg |
| **TBS45** | Neg | Pos | Neg | Neg | Neg | Neg | Neg |
| **TBS55** | Pos | Pos | Neg | Neg | Neg | Neg | Neg |
| **TBS109** | Pos | Pos | Neg | Neg | Pos | Pos | Neg |

Pos: Positive; Neg: Negative.

nitrogen gas (N$_2$ gas) is a distinct quality found only in TBS35, indicating its potential role in nitrogen fixation or denitrification procedures. The remaining tests conducted for arginine dihydrolase (ADH), ornithine decarboxylase (ODC), and lysine decarboxylase (LDC) mainly produced negative outcomes among various species. However, one exception, TBS35, displayed positive outcomes for ODC and LDC. Table 7 reveals the wide range of metabolic abilities and enzyme functions found in the thermophilic *Bacillus* species that have been isolated.

## An analysis of DNA concentration, 16s rRNA gene length, and G+C content

Table 9 represents a diverse set of bacterial samples, each characterized by varying DNA extraction concentrations, 16s rRNA gene lengths, and G+C contents. For instance, *B.*

**Table 9. Comparative analysis of enzymatic activities and stress tolerance in thermophilic *Bacillus* isolates.**

| Thermophilic *Bacillus* species | Enzymatic activities and stress tolerance | | | | | | | | |
|---|---|---|---|---|---|---|---|---|---|
| | 7% NaCl | IND | VP | NO2 | ONPG | N$_2$ gas | ADH | ODC | LDC |
| *B. stearothermophilus* | Neg | Neg | Neg | Pos | Neg | Neg | Neg | Neg | Neg |
| **TBS4** | Neg | Pos | Neg | Pos | Pos | Neg | Neg | Neg | Neg |
| **TBS5** | Neg | Pos | Pos | Pos | Pos | Neg | Neg | Neg | Neg |
| **TBS6** | Neg | Neg | Pos | Pos | Pos | Neg | Neg | Neg | Neg |
| **TBS25** | Neg | Neg | Neg | Pos | Pos | Neg | Neg | Neg | Neg |
| **TBS26** | Neg | Pos | Neg | Pos | Pos | Neg | Neg | Neg | Neg |
| **TBS34** | Neg | Neg | Neg | Pos | Pos | Neg | Neg | Neg | Neg |
| **TBS35** | Pos | Neg | Neg | Neg | Pos | Pos | Neg | Neg | Pos |
| **TBS40** | Pos | Neg | Neg | Neg | Neg | Neg | Neg | Neg | Neg |
| **TBS45** | Neg | Neg | Neg | Neg | Neg | Neg | Neg | Neg | Neg |
| **TBS55** | Neg | Pos | Pos | Pos | Pos | Neg | Neg | Neg | Neg |
| **TBS109** | Pos | Neg | Pos | Pos | Pos | Neg | Neg | Neg | Neg |

Indole production (IND); Nitrate reduction(NO$_2$); O-nitrophenyl-β-D-galactopyranoside (ONPG); Voges-Proskauer test (VP); nitrogen gas (N$_2$ gas); arginine dihydrolase (ADH); ornithine decarboxylase (ODC); lysine decarboxylase (LDC).

*stearothermophilus* showcased a moderately high DNA concentration and G+C content, aligning with its characteristics as a thermophilic bacterium. Similarly, TBS4, TBS6, TBS40, and TBS45 share a consistent 16s rRNA gene length but display distinct DNA concentrations and G+C contents, hinting at closely related species with varied genomic features. Interestingly, TBS5 and TBS25 exhibited identical 16s rRNA gene lengths and G+C contents, suggesting they might be the same species or closely related despite their different DNA concentrations. In contrast, TBS26 had the highest G+C content and a longer 16s rRNA gene, indicating unique adaptations or genomic traits. The highest DNA concentration was observed in TBS34, coupled with the lowest G+C content, pointing to a unique bacterial strain with specific adaptation strategies. TBS55, with a high DNA concentration but moderate G+C content and a shorter 16s rRNA gene, further illustrates the genomic diversity within these samples. Lastly, TBS109 descended into a moderate range for DNA concentration and G+C content, with a slightly longer 16s rRNA gene. This data assortment reflected the vast array of environmental adaptations and evolutionary histories represented within these bacterial samples, highlighting the complexity and diversity of bacterial genomics.

## Discussion

This study's comprehensive analysis of isolated thermophilic *Bacillus* species reveals significant insights into their physiological and genetic diversity, with implications for understanding their roles in various environments and potential applications in biotechnology and industry.

Our findings in Tables 2 and 10 demonstrate that these *Bacillus* species have adapted to various temperatures and pH conditions. The universal upper limit of growth at 73°C across all species underlines a potential threshold for thermophilic bacterial survival and function. The variation in optimal growth temperatures, with species like *B. stearothermophilus* thriving at 56°C while others like TBS5 and TBS35 prefer higher temperatures, reflects their ecological adaptability [7,8,19]. The ability of TBS35 and TBS40 to grow at a lower minimum temperature of 37°C could indicate a broader ecological niche or specific adaptive mechanisms [20–22].

The diversity in pH tolerance, especially the unique adaptation of TBS34, TBS35, and TBS40 to thrive in highly alkaline conditions (pH 11), suggests that these species have evolved specialized mechanisms to maintain cellular homeostasis under extreme pH conditions, aligning with findings by [7,8,19]. This adaptability could be exploited in industrial processes where extreme pH conditions are prevalent.

**Table 10. Comparative analysis of isolated thermophilic *Bacillus* species genomic characteristics: DNA concentration, 16s rRNA gene length, and G+C content.**

| Thermophilic *Bacillus* species | DNA concentration (ng/µL) | 16s rRNA gene (bp) | G+C (%) |
|---|---|---|---|
| *B. stearothermophilus* | 44.5 | 1400 | 52 |
| **TBS4** | 48.25 | 1200 | 50 |
| **TBS5** | 46.78 | 1550 | 61 |
| **TBS6** | 26.58 | 1200 | 48 |
| **TBS25** | 19.04 | 1550 | 61 |
| **TBS26** | 22.86 | 1600 | 64 |
| **TBS34** | 123.4 | 1100 | 35 |
| **TBS35** | 7.76 | 1200 | 40 |
| **TBS40** | 43.35 | 1200 | 42 |
| **TBS45** | 8.39 | 1350 | 49 |
| **TBS55** | 144.3 | 1100 | 45 |
| **TBS109** | 20.4 | 1500 | 52 |

Examining Table 3 more closely, which specifically examines the oxygen needs of the *Bacillus* species, uncovers several preferences for aerobic, anaerobic, and facultative aerobic growth. This diversity in oxygen requirements demonstrates the inherent adaptability of these bacteria to various ecological settings. Their ability to efficiently utilize different oxygen levels enables them to thrive in environments with fluctuating oxygen concentrations, highlighting their resilient survival tactics [19,22]. Our study observed that the isolated *Bacillus* species exhibited a significant variation in their oxygen requirements. This finding contrasts with the research conducted by [19,20,21], which found a more consistent preference for oxygen among the thermophilic bacteria studied. These differences suggest that there may be variations in environmental adaptation among different strains or species of bacteria [18,19,20].

The colony morphology, broth growth, and hemolytic properties detailed in Table 4 reveal significant diversity. The variations in colony color, shape, and hemolytic properties could indicate different metabolic pathways and virulence factors, essential for understanding their ecological interactions and potential pathogenicity [23–26].

The morphological and cellular diversity highlighted in Table 5, including cell shape and spore position, aligns with the evolutionary adaptations discussed by Li et al. [27]. The universal motility observed among these species suggests advantages in environmental navigation, supporting findings by Xiaomei et al. [28] on bacterial motility in bioremediation.

Table 6 emphasizes carbohydrate fermentation, revealing different *Bacillus* species' varied and specialized metabolism. While they commonly can ferment glucose, they also exhibit a selective preference for other types of carbohydrates. This selective behavior in using substrates highlights the flexibility of metabolic pathways and demonstrates their capability to efficiently extract energy from the resources at hand [8,23,29]. In addition, our investigation has uncovered diverse metabolic abilities and specific preferences for different substances within *Bacillus* species. This discovery aligns with the findings of [20,25,30], who also studied the fermentation of carbohydrates by heat-loving bacteria. Their research emphasized comparable tendencies of favoring glucose and selectively using other carbohydrates, further supporting that these organisms possess a shared characteristic of adaptability in their metabolism [20,25,30].

Upon reviewing Table 7, we discovered a wealth of data concerning the synthesis of enzymes and the breakdown of substrates. The fact that multiple species produce catalase and oxidase enzymes suggests that they all employ similar methods to cope with oxidative stress and can use different sources of electrons. At the same time, the specific breakdown of substances such as starch and gelatin highlights the inherent metabolic diversity and specialization of these bacteria [5,23,31]. Our *Bacillus* isolates demonstrate similar enzyme production and substrate hydrolysis abilities, as shown in studies conducted by [1,5,9], where various enzymatic activities were discovered in thermophilic bacteria. However, our isolates exhibited a greater diversity and specificity in enzyme production, indicating a wider range of metabolic capabilities than previous findings [5,21].

Exploring Table 8 reveals further insights into the metabolic abilities and adaptability of the *Bacillus* species. The differences observed in their ability to tolerate salt, produce indole and acetone, reduce nitrate, and break down amino acids provide a glimpse into the versatility of these species in different environments. These discoveries indicate that they may play an important role in nutrient recycling and interaction with their surroundings, highlighting their ecological importance even more [8,32].

Finally, the analysis of DNA concentration, 16s rRNA gene length, and G+C content in Table 9 offers profound insights into the genomic diversity and evolutionary trajectories of the *Bacillus* species studied. These genetic markers are crucial indicators, revealing a spectrum of adaptive strategies and evolutionary pathways [33]. For instance, variations in 16s rRNA gene lengths shed light on the phylogenetic relationships among the species, suggesting unique

evolutionary histories. Meanwhile, differences in G+C content may reflect adaptations to specific environmental pressures, such as temperature tolerance, which is particularly relevant for thermophilic bacteria [34]. This correlation between genetic characteristics and adaptability highlights the immense potential of these species in biotechnological applications [35]. The observed genetic diversity underscores the species' ability to thrive in diverse environments and points to their utility in industrial processes.

Further, by correlating these genetic markers with phenotypic traits, a more comprehensive understanding of how genetic variations manifest in physiological characteristics can be achieved [36]. Placing these findings within the broader context of bacterial genomics underscores their significance, paving the way for future research, such as functional genomics studies, to explore the roles of specific genes in adaptability and the potential exploitation of these strains in various biotechnological arenas [37]. This comprehensive analysis, therefore, illuminates the intricate interplay between genetic composition, environmental adaptability, and evolutionary history in *Bacillus* species, opening new avenues for scientific exploration and practical applications.

Our study not only sheds light on the ecological roles of these bacteria but also opens up potential applications in various biotechnological fields. The adaptability of these species to a wide range of temperatures, pH levels, and oxygen conditions, as detailed in our findings, underscores their potential as robust candidates for bioremediation, biofuel production, and pharmaceutical manufacturing. This study extends beyond academic curiosity, offering practical solutions to industrial challenges and contributing to environmental sustainability. The diversity in metabolic and enzymatic capabilities, as observed, suggests these *Bacillus* strains could be pivotal in developing sustainable industrial practices. Moreover, our findings provide a foundation for future research, including engineering bacteria for specific applications, further exploring their roles in diverse environments, and developing innovative biotechnological tools. The ecological versatility of these bacteria, highlighted in our study, enriches our understanding of bacterial ecology, and has tangible implications in addressing global challenges like climate change and resource scarcity. In conclusion, this comprehensive analysis not only deepens our understanding of *Bacillus* species but also bridges the gap between basic science and practical applications, highlighting the significant impact of our research in both scientific and industrial realms.

## Conclusions

Overall, this study provides a comprehensive overview of thermophilic *Bacillus* species' physiological, morphological, and genetic diversity. These findings have significant implications for understanding their roles in natural ecosystems and potential applications in industrial processes, bioremediation, and biotechnology. The diverse adaptations to temperature, pH, oxygen availability, and nutrient sources underscore the ecological versatility of these bacteria and their potential as robust candidates for various biotechnological applications.

## Supporting information

**S1 Fig. Cell morphology of TBS4 using Gram-stain, spores were subterminal.** Bacteria cell morphology (A) at 56˚C; and (B) at 73˚C.
(DOCX)

**S2 Fig. Cell morphology of TBS5 using Gram-stain, spores were central and subterminal.** Bacteria cell morphology (A) at 50˚C; and (B) at 73˚C.
(DOCX)

**S3 Fig. Cell morphology of TBS6 using Gram-stain, spores were subterminal.** Bacteria cell morphology (A) at 51˚C; and (B) bacteria grew at 73˚C.
(DOCX)

**S4 Fig. Cell morphology of TBS25 using Gram-stain, spores were subterminal.** Bacteria cell morphology (A) at 54˚C; and (B) bacteria grew at 73˚C.
(DOCX)

**S5 Fig. Cell morphology of TBS26 using Gram-stain, spores were central and subterminal.** Bacteria cell morphology (A) at 51˚C; and (B) bacteria grew at 73˚C.
(DOCX)

**S6 Fig. Cell morphology of TBS34 using Gram-stain, spores were subterminal.** Bacteria cell morphology (A) at 54˚C; and (B) bacteria grew at 73˚C.
(DOCX)

**S7 Fig. Cell morphology of TBS35 using Gram-stain, spores were subterminal.** Bacteria cell morphology (A) at 60˚C; and (B) bacteria grew at 73˚C.
(DOCX)

**S8 Fig. Cell morphology of TBS40 using Gram-stain, spores were subterminal.** Bacteria cell morphology (A) at 60˚C; and (B) bacteria grew at 73˚C.
(DOCX)

**S9 Fig. Cell morphology of TBS45 using Gram-stain, spores were terminal.** Bacteria cell morphology (A) at 55˚C; and (B) bacteria grew at 73˚C.
(DOCX)

**S10 Fig. Cell morphology of TBS55 using Gram-stain, spores were central.** Bacteria cell morphology (A) at 52˚C; and (B) bacteria grew at 73˚C.
(DOCX)

**S1 Table. Cell morphology of isolated thermophilic *Bacillus* strains at optimum and maximum temperatures.**
(DOCX)

## Author Contributions

**Conceptualization:** Hazem Aqel, Husni Farah, Afnan Al-Hunaiti.

**Data curation:** Hazem Aqel, Husni Farah, Afnan Al-Hunaiti.

**Formal analysis:** Hazem Aqel, Husni Farah.

**Investigation:** Hazem Aqel, Husni Farah.

**Methodology:** Hazem Aqel, Husni Farah, Afnan Al-Hunaiti.

**Project administration:** Hazem Aqel.

**Resources:** Hazem Aqel, Husni Farah.

**Software:** Afnan Al-Hunaiti.

**Supervision:** Hazem Aqel.

**Validation:** Hazem Aqel, Afnan Al-Hunaiti.

**Visualization:** Hazem Aqel, Afnan Al-Hunaiti.

**Writing – original draft:** Hazem Aqel.

**Writing – review & editing:** Husni Farah, Afnan Al-Hunaiti.

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
