## [Decision Letter · Decision Letter 0]

15 Nov 2023

PONE-D-23-31383Ecological versatility and biotechnological promise: Comprehensive characterization of isolated thermophilic Bacillus StrainsPLOS ONE

Dear Dr. Aqel,

Thank you for submitting your manuscript to PLOS ONE. After careful consideration, we feel that it has merit but does not fully meet PLOS ONE’s publication criteria as it currently stands. Therefore, we invite you to submit a revised version of the manuscript that addresses the points raised during the review process.

The submitted manuscript needs a major revision. As has been pointed out by one of the reviewers, at present without molecular characterization identification of the bacterial species is unacceptable. The author's therefore may please be communicated that in view of the reviewer's comments they need to revise the manuscript before it is considered for publication. Please submit your revised manuscript by Dec 30 2023 11:59PM. If you will need more time than this to complete your revisions, please reply to this message or contact the journal office at plosone@plos.org. Please include the following items when submitting your revised manuscript:A rebuttal letter that responds to each point raised by the academic editor and reviewer(s). You should upload this letter as a separate file labeled 'Response to Reviewers'.A marked-up copy of your manuscript that highlights changes made to the original version. You should upload this as a separate file labeled 'Revised Manuscript with Track Changes'.An unmarked version of your revised paper without tracked changes. You should upload this as a separate file labeled 'Manuscript'.

We look forward to receiving your revised manuscript.

Kind regards,

Faiz ul-Hassan Nasim, PhD

Academic Editor

PLOS ONE

Journal Requirements:

Additional Editor Comments:

Please see the comments of the reviewers and proceed accordingly.

Reviewers' comments:

Reviewer's Responses to Questions

**Comments to the Author**

1. Is the manuscript technically sound, and do the data support the conclusions?

Reviewer #1: Yes

Reviewer #2: Partly

2. Has the statistical analysis been performed appropriately and rigorously? 

Reviewer #1: N/A

Reviewer #2: N/A

3. Have the authors made all data underlying the findings in their manuscript fully available?

Reviewer #1: Yes

Reviewer #2: Yes

4. Is the manuscript presented in an intelligible fashion and written in standard English?

Reviewer #1: Yes

Reviewer #2: Yes

5. Review Comments to the Author

Reviewer #1: In the abstract section, the findings are verbally expressed without substantiating the same with data. This is not a standard practice. The authors should have briefly described with data (in the abstract section) how the Bacillus strains are adapted to extreme environmental conditions, and what ecological factors drive these adaptabilities. The objectives of the work should be described with clarity in the abstract section.

Reviewer #2: The manuscript entitled "Ecological versatility and biotechnological promise: Comprehensive characterization of

isolated thermophilic Bacillus Strains" highlights the isolation and biochemical characterization of Bacillus species from host springs of Jordan.

The manuscript is written well with comprehensive methodology supported by results and discussion sections. The research providing the basic data about various isolates of Bacillus bacteria. It is already understood about these thermoduric Bacillus species that they have same biochemical properties enlisted by the researchers in this manuscript. Therefore, there are few concerns about the study design and usefulness of the research by scientific community.

1. What is the significance of the current study to the wider scientific community?

2. The authors unable to demonstrate the use of provided data in terms of applied research.

3. Another important aspect is the missing of molecular characterization as biochemical analysis alone is not sufficient in current era to establish the basic features of the bacterial species.

6. PLOS authors have the option to publish the peer review history of their article (what does this mean?). If published, this will include your full peer review and any attached files.

Reviewer #1: **Yes: **Dr. Abdullah Salim Khan

Reviewer #2: No

---

## [Author Response · Author response to Decision Letter 0]

6 Dec 2023

Reviewer #1,

Thank you for your comments and suggestions on our manuscript. Based on your feedback, we have revised and corrected the abstract to better reflect the scope and findings of our study. The modified abstract now clearly outlines the focus on the adaptability and physiological diversity of isolated thermophilic Bacillus species, emphasizing their ecological roles and potential industrial applications.

We have expanded our investigation of these species' capacity to thrive in extreme conditions by detailing their environmental tolerances and adaptations at the metabolic and genetic levels. The abstract now explicitly states the primary objective of evaluating the suitability of these species for biotechnological applications, considering their resilience in harsh environments.

Furthermore, we have included a more comprehensive description of our comparative analysis of the environmental adaptability parameters for various Bacillus species. This encompasses growth temperature ranges, pH tolerance, oxygen requirements, carbohydrate fermentation patterns, colony morphology, enzymatic activities, and genetic properties. We have also specified the controlled laboratory experiments that provided the data for our analysis.

Key findings have been highlighted more prominently, such as the species' ability to endure temperatures as high as 73°C, their preference for slightly alkaline conditions, and their classification into aerobic, anaerobic, and facultative aerobic categories based on oxygen requirement tests. Genetic analysis and enzymatic profiles have also been elaborated to showcase the adaptability and diversity of these species under various environmental conditions.

Overall, the revised abstract more accurately and comprehensively conveys the significance of our findings and the potential of thermophilic Bacillus species in biotechnological applications. We believe these changes address your concerns and enhance the clarity and impact of the abstract.

Thank you for your constructive feedback, which has been invaluable in improving our manuscript.

Sincerely,

Hazem

Reviewer #2,

Thank you for your invaluable feedback and insights. We have addressed the concerns you raised in your review, particularly regarding the molecular characterization of the Bacillus strains and the significance and usefulness of our study.

Molecular Characterization:

In response to your suggestion, we have conducted additional molecular analyses to characterize the bacterial species more definitively. This involved using %G+C content and PCR-based methods, which have enriched our understanding of the identity of the Bacillus strains. The results of these analyses are now included in the revised manuscript, providing a more robust molecular characterization of the isolated strains.

Significance and usefulness of the study:

We have clarified the significance of our study to the wider scientific community. Our findings contribute new knowledge in the field of microbiology, particularly in understanding the adaptability and physiological diversity of thermophilic Bacillus species. The potential applications of these findings in biotechnology and other relevant fields have been emphasized, including the use of these bacteria in bioremediation, biofuel production, and pharmaceutical manufacturing.

Practical Applications of the data:

The revised manuscript now includes a detailed explanation of how our research findings can be applied in various biotechnological applications. We elucidate the potential uses of these Bacillus strains, considering their ability to withstand extreme conditions, their diverse metabolic capabilities, and the genetic adaptability they exhibit. This section aims to bridge the gap between our fundamental research and its practical applications, highlighting the industrial relevance of our findings.

Results and Discussion:

The revised Results and Discussion sections incorporate a comprehensive analysis of DNA concentration, 16s rRNA gene length, and G+C content. These sections provide insights into the genetic diversity of the Bacillus species studied, emphasizing their adaptability and potential in various applications. The discussion now includes a comparative analysis of the environmental adaptability parameters, providing a deeper understanding of the physiological and genetic diversity of these species.

Conclusions:

The conclusions have been updated to reflect the comprehensive nature of our study. They emphasize the implications of our findings in understanding the roles of thermophilic Bacillus species in natural ecosystems and their potential applications in industrial processes, bioremediation, and biotechnology. The diversity in adaptations observed in these bacteria underscores their ecological versatility and potential as robust candidates for a variety of biotechnological applications.

We believe these revisions and additions address your concerns and significantly enhance the quality and impact of our manuscript. Your feedback has been instrumental in guiding these improvements, and we are grateful for your thorough and constructive review.

Sincerely,

Hazem

Dear Dr. Nasim,

Subject: Submission of Revised Manuscript to PLOS ONE

I am writing to submit the revised version of our manuscript [Ecological Versatility and Biotechnological Promise: Comprehensive Characterization of The Isolated Thermophilic Bacillus Strains] for consideration in PLOS ONE. We have carefully reviewed and addressed the comments and suggestions provided by the reviewers. Enclosed with this letter are the following documents as per the journal's requirements:

1. Marked-up copy of the manuscript: This version highlights all the changes made in response to the reviewers' comments. Changes include textual revisions, additional data, and methodological clarifications. We have used track changes for easy identification of the modifications.

2. Unmarked version of the revised paper: This is a clean copy of the revised manuscript, incorporating all the changes without the markup. It is formatted according to the guidelines of PLOS ONE and is ready for publication.

Dear [Reviewer #1] and [Reviewer #2],

Thank you for your constructive comments and suggestions regarding our manuscript [Ecological Versatility and Biotechnological Promise: Comprehensive Characterization of The Isolated Thermophilic Bacillus Strains]. Your feedback has greatly contributed to enhancing the quality and clarity of our research. Below, we address each point raised and explain how we have incorporated your feedback into the revised manuscript.

Response to Reviewer #1

1. Point about data supporting findings: We have revised the abstract and relevant sections to include specific data points and summaries that support our claims. These changes provide clearer evidence of the adaptability and potential applications of Bacillus strains.

2. Clarification of objectives: The objectives of the study have been explicitly stated in the abstract and introduction, providing a clear outline of our research aims.

Respond to Reviewer #2

1. Significance of the study: We have expanded the discussion section to elaborate on how our findings contribute new knowledge to the scientific community, especially in the context of biotechnology and microbial ecology.

2. Practical applications: We have included a detailed explanation of the potential biotechnological applications of the Bacillus strains in applied research, emphasizing their relevance in industrial processes and environmental conservation.

General Formatting and submission guidelines

We have ensured that the manuscript adheres to the PLOS ONE formatting and submission guidelines. All figures and tables are formatted as per the journal's requirements, and references have been updated accordingly.

We believe that the revisions made have significantly strengthened the manuscript, making it a valuable contribution to PLOS ONE. We appreciate the opportunity to revise our manuscript and thank the reviewers and the editor for their insightful and constructive feedback.

Sincerely,

Hazem

---

## [Editor Report · Decision Letter 1]

2 Jan 2024

Ecological versatility and biotechnological promise: Comprehensive characterization of the isolated thermophilic Bacillus Strains

PONE-D-23-31383R1

Dear Dr. Aqel,

We’re pleased to inform you that your manuscript has been judged scientifically suitable for publication and will be formally accepted for publication once it meets all outstanding technical requirements.

Kind regards,

Faiz ul-Hassan Nasim, PhD

Academic Editor

PLOS ONE

Additional Editor Comments (optional):

Accepted for publication.
---

## [Editor Report · Acceptance letter]

24 Jan 2024

PONE-D-23-31383R1 

PLOS ONE

Dear Dr. Aqel, 

I'm pleased to inform you that your manuscript has been deemed suitable for publication in PLOS ONE. Congratulations! Your manuscript is now being handed over to our production team.

Kind regards, 

on behalf of

Dr. Faiz ul-Hassan Nasim 

Academic Editor

PLOS ONE